Comparative transcriptomics reveals the difference in early endosperm development between maize with different amylose contents

Qu Jianzhou 1 2
Xu Shutu 1 2
Tian Xiaokang 1 2
Li Ting 1 2
Wang Licheng 1 2
Zhong Yuyue 1 2
Xue Jiquan xjq2934@163.com 1 2
Guo Dongwei gdwei@nwsuaf.edu.cn gdwei1973@126.com 1 2
1 The Key Laboratory of Biology and Genetics Improvement of Maize in Arid Area of Northwest Region, Ministry of Agriculture and Rural Affairs, College of Agronomy, Northwest A&F University , Yangling , Shaanxi , China
2 Maize Engineering Technology Research Centre of Shaanxi Province , Yangling , Shaanxi , China
Misra Biswapriya
Electronic publication date: 2019 Aug 28
Publication date: 2019
Volume: 7
Electronic Location ID: e7528
Received 2019 Mar 8; Accepted 2019 Jul 22
Copyright: ©2019 Qu et al.
Copyright year: 2019
Copyright holder: Qu et al.
License: This is an open access article distributed under the terms of the Creative Commons Attribution License, which permits unrestricted use, distribution, reproduction and adaptation in any medium and for any purpose provided that it is properly attributed. For attribution, the original author(s), title, publication source (PeerJ) and either DOI or URL of the article must be cited.
License URL: https://creativecommons.org/licenses/by/4.0/

Keywords: Gene expression, Maize, Starch metabolism, RNA-sequence, Endosperm

Funding: Shaanxi Province Science and Technology Innovation Coordination Project 2015KTZDNY01-01-01 Natural Key Research and Development Program of China 2017YFD0300304 This work was supported by the Shaanxi Province Science and Technology Innovation Coordination Project (2015KTZDNY01-01-01) and the Natural Key Research and Development Program of China (2017YFD0300304). The funders had no role in study design, data collection and analysis, decision to publish, or preparation of the manuscript.

==============================
In seeds, the endosperm is a crucial organ that plays vital roles in supporting embryo development and determining seed weight and quality. Starch is the predominant storage carbohydrate of the endosperm and accounts for ∼70% of the mature maize kernel weight. Nonetheless, because starch biosynthesis is a complex process that is orchestrated by multiple enzymes, the gene regulatory networks of starch biosynthesis, particularly amylose and amylopectin biosynthesis, have not been fully elucidated. Here, through high-throughput RNA sequencing, we developed a temporal transcriptome atlas of the endosperms of high-amylose maize and common maize at 5-, 10-, 15- and 20-day after pollination and found that 21,986 genes are involved in the programming of the high-amylose and common maize endosperm. A coexpression analysis identified multiple sequentially expressed gene sets that are closely correlated with cellular and metabolic programmes and provided valuable insight into the dynamic reprogramming of the transcriptome in common and high-amylose maize. In addition, a number of genes and transcription factors were found to be strongly linked to starch synthesis, which might help elucidate the key mechanisms and regulatory networks underlying amylose and amylopectin biosynthesis. This study will aid the understanding of the spatiotemporal patterns and genetic regulation of endosperm development in different types of maize and provide valuable genetic information for the breeding of starch varieties with different contents.

Introduction

Maize (Zea mays) is one of the most important food and feed crop sources worldwide. The mature maize seed consists of the endosperm, embryo, and seed coat, and the endosperm occupies ∼85% of the seed volume and is a critical tissue influencing the maize grain weight (Lopes & Larkins, 1993; Rousseau et al., 2015). The mature endosperm contains large amounts of carbohydrates and proteins that provide nutrients to the developing embryo and seedling, and starch is the main type of carbohydrate in the maize endosperm, accounting for ∼70% of the endosperm volume. Additionally, maize starch is one of the best-quality starches, and 80% of the world’s starch is obtained from maize (Zhang et al., 2013).

The development of the maize endosperm starts after double fertilization, and the process can be classified as a nuclear process. A time-series analysis of maize endosperm development has identified four phases: syncytial, cellularization, differentiation and death (Berger, 1999). The development of the maize endosperm is initiated by successive divisions of the triploid zygotic nucleus without cytokinesis, and this process, which occurs 0∼3 days after pollination (DAP), creates a coenocyte that contains 128 to 512 nuclei (Olsen, 2004; Leroux et al., 2014; Larkins, 2017). Starting at approximately 4∼6 DAP, the cellularization phase of endosperm development can include differentiation into four recognizable cell types: basal endosperm transfer layer cells, aleurone cells, embryo-surrounding region cells and central starchy endosperm cells (Olsen, 2004). At approximately 8 DAP, subaleurone cells, conducting zone cells and basal intermediate zone cells emerge and become cytologically identifiable (Larkins, 2017). Subsequently, at approximately 8∼12 DAP, the endosperm undergoes rapid cell divisions, and the cells in the central portion of the endosperm gradually switch from a mitotic cell cycle to endoreduplication and start accumulating substantial amounts of starch and storage proteins (Sabelli & Larkins, 2009; Becraft & Gutierrez-Marcos, 2012). At 20∼25 DAP, the process of cell division enters a late stage, and the outer two cell layers of the endosperm develop into the aleurone and the subaleurone (Olsen, 2004). During this late period of endosperm development, the maize endosperm undergoes maturation and desiccation, and cell death results in dormancy.

Gene expression is key for revealing the mechanisms underlying the development of the maize endosperm. A series of genetic experiments have identified a number of genes that play important roles in the regulation of endosperm development. Examples of these genes include maize GLOBBY1 and H3K27 methyltransferase because mutations in these genes induce abnormal endosperm cellularization; moreover, the GLOBBY1 gene also regulates early cell proliferation and differentiation of the endosperm, whereas H3K27 methyltransferase increases coenocytic proliferation (Costa et al., 2003; Gehring & Satyaki, 2017). Additionally, the mutation or deletion of APETALA2 results in delayed endosperm cellularization and an increased seed size, whereas a mutation in AGAMOUS-LIKE62 results in precocious endosperm cellularization and a reduced seed size (Kang et al., 2008; Orozco-Arroyo et al., 2015).

The accumulation and synthesis of reserve metabolites and endosperm development are inextricably linked. During endosperm development, aleurone cells form a major nutritive layer that stores proteins, lipids, non-starch carbohydrates, and mineral nutrients. In addition, even if all endosperm cells undergo programmed cell death during endosperm maturation, the aleurone cells remain alive (Becraft & Gutierrez-Marcos, 2012). Genetic studies have revealed that mutation of the naked endosperm (nkd) gene not only affects the fate and differentiation of aleurone cells but also alters the accumulation and synthesis of reserve metabolites by regulating widespread gene expression (Gontarek et al., 2016). For example, opaque-2 and prolamin box-binding factor are regulatory genes of naked endosperm1 and naked endosperm2, which play important roles in controlling multiple classes of zeins and show downregulated expression in aleurone cells and nkd-mutant cells (Gontarek et al., 2016). The mutation of opaque-2 results in a starchy endosperm, but high expression of 27-kD γ-zein can convert the starchy endosperm to a vitreous phenotype (Wu & Messing, 2010). Furthermore, opaque-2 and prolamin box-binding factors are involved in the regulation of carbon and nitrogen metabolism, pathogens and stress and alter the lysine content during endosperm development (Li et al., 2015; Morton et al., 2016). These factors also indirectly regulate the transcript and encoded protein levels of SSIIa and SBEI in starch synthesis (Zhang et al., 2016).

In higher plants, the starch biosynthesis pathway is orchestrated by multiple enzymes, which mainly include ADP-glucose pyrophosphorylase (AGPase), granule-bound starch synthase (GBSS), soluble starch synthase (SS), starch branching enzyme (SBE), and starch debranching enzyme (DBE). Starch consists of two types of polysaccharides, namely, amylose and amylopectin, and the ratio of amylose to amylopectin determines the suitability of starch for particular end-uses, such as gelatinisation/gelation characteristics, solubility, the formation of resistant starch, cooking, textural characteristics, and digestibility or nutritional values (Jane et al., 1999; Varavinit et al., 2003; Li et al., 2008; Witt, Gidley & Gilbert, 2010; Zhu et al., 2011). Thus, the amylose content is frequently used to predict the properties of starch, and starch with a high amylose content is optimal for the manufacturing of photodissociative plastics and helps control serious “white pollution” (Jiang et al., 2010; Jiang et al., 2013). Many researchers have reported that a mutation in amylose extender1 (ae1) yields maize kernels with a higher amount of amylose than the levels found in nonmutant kernels (Fisher et al., 1996; Kim et al., 1998). Additionally, GBSSI is a key enzyme for amylose synthesis, and the activity of GBSSI is decreased (∼5–95%) during amylose synthesis in wx1-mutant maize, resulting in a higher level of amylopectin in the maize endosperm and pollen (Sprague, Brimhall & Nixon, 1943; Nelson & Rines, 1962). These two mutants are currently used to create either high- or low-amylose maize and thus yield corn with altered starch properties and utilities.

However, the gene network that links endosperm development with starch synthesis remains unclear. The development of high-throughput technology has resulted in a massive increase in the amount of RNA-Seq data available, and these data are highly reproducible, with few systematic discrepancies among technical replicates (Marioni et al., 2008). Moreover, coexpression analyses based on these large-scale data have been performed for the screening and identification of novel genes that might be involved in metabolic pathways (Saito, Hirai & Yonekura-Sakakibara, 2008; Higashi & Saito, 2013). These techniques and methods provide a novel approach for revealing the relationship between metabolism and development. Here, we utilized next-generation sequencing technology to profile the gene expression and function of the endosperms of maize with different amylose contents at 5, 10, 15 and 20 DAP. The results will provide valuable insight into the underlying mechanisms that determine the starch content of seeds, particularly the relationship between starch metabolism and endosperm development.

Materials & Methods

Plant materials and sequencing

The common maize (Zea mays. L) hybrid Shandan 609 (SD609) and the high-amylose maize hybrid high-amylose 68 (HS68) were grown under field conditions at a density of approximately 67,500 plants ha−1 in the summer of 2013 in Yangling, Shaanxi Province, China. The ears were bagged prior to silk emergence and then subjected to manual self-pollination. Three biological replicates of the ears were harvested at 5, 10, 15 and 20 DAP. To ensure the integrity and specificity of the endosperm tissues and the absence of contamination by any other tissues, all slightly damaged endosperm tissues were discarded, and only complete tissues were frozen immediately in liquid nitrogen and stored at −80 °C prior to RNA extraction.

The total RNA of three biological replicates of the endosperm was extracted separately from each sample using the TRIzol-A+ reagent (Invitrogen) following the manufacturer’s instructions. RNA sequencing libraries were constructed according to the standard Illumina protocol and were sequenced using the Illumina HiSeq™ 2000 platform to generate 2 ×100-nucleotide paired-end reads (Qu et al., 2016).

The sequence data obtained in this study have been deposited in the National Center for Biotechnology Information Sequence Read Archive (http://www.ncbi.nlm.nih.gov/sra) under the accession numbers SRP065059 and SRP149609.

Read mapping and expression analysis

RNA-Seq reads from each sample were aligned to the maize B73 reference genome (ZmB73_RefGen_v2; http://www.maizesequence.org/) using the Tophat2 (v2.09; http://ccb.jhu.edu/software/tophat/index.shtml) programme with Bowtie2 read alignment software (Trapnell, Pachter & Salzberg, 2009; Langmead et al., 2009). The aligned reads obtained after mapping using Tophat2 were subjected to transcript assembly using Cufflinks software (version 2.1.0; http://cole-trapnell-lab.github.io/cufflinks). The expression level was calculated in fragments per kilobase of transcript per million fragments mapped (FPKM) as described previously (Qu et al., 2016). The differentially expressed genes (DEGs) were identified through pairwise comparison using EBSeq software, and the P-values were adjusted using the Benjamini–Hochberg procedure to determine the false discovery rate (FDR) (Benjamini & Hochberg, 1995; Robinson, McCarthy & Smyth, 2010; Leng et al., 2013). Only the genes with FPKM results that met the criteria of FDR <0.01 and fold change  ≥2 or ≤0.5 between the two conditions were deemed to be significantly differentially expressed.

Gene coexpression network analysis

A gene coexpression network analysis was performed using the WGCNA R package (version1.63; https://horvath.genetics.ucla.edu/html/CoexpressionNetwork/Rpackages/WGCNA/) (Langfelder & Horvath, 2008). We selected all DEGs for the network analysis based on their variance across different samples. We subsequently calculated the Pearson correlation coefficients for each gene-gene comparison and then constructed an adjacency matrix of the connection strengths. The power β was optimized to 6 to adjust the scale-free property of the coexpression network and the sparsity of the connections between genes. Modules were defined as sets of genes showing high topological overlap. An adjacency matrix was used to calculate the topographical overlap matrix (TOM), and a TOM-based dissimilarity measure (1–TOM) was used for hierarchical clustering. The cutreeDynamic function was used to define the gene modules corresponding to the branches of the resulting dendrogram. Highly similar clusters were merged in the network using the mergeCloseModules function with a cutHeight value of 0.25. Each module was assigned a unique number. The NetworkAnalyzer plugin available in Cytoscape was used to calculate the relevant network parameters, such as the degree of connection (Assenov et al., 2008), which measures the number of incoming and outgoing edges of a node. A higher degree of connection indicates that the node has a higher number of edges, and thus, this property can be used to identify whether a gene is a major hub of the network and thus potentially a major regulatory gene in a pathway. The top 5% of genes in each module were identified as hub genes. The modules were visualized using Cytoscape (version 3.6.1; http://www.cytoscape.org/download.php), and the layout was set based on an edge-weighted spring-embedded layout (Shannon et al., 2003).

Statistical analyses

Functional enrichment analyses were performed using the KEGG database (https://www.genome.jp/kegg/) and agriGO (http://bioinfo.cau.edu.cn/agriGO/analysis.php; version 2.0) (Du et al., 2010). A gene ontology (GO) term was considered significantly enriched if the adjusted p-value was lower than 0.05 compared with the genome-wide background. The K-mean clustering method, Euclidean distance and Calinski-Harabasz index were used to identify the gene expression patterns (Calinski & Harabasz, 1974). Promoter sequences of the core genes in starch synthesis were obtained using the Ensembl Plants database (http://plants.ensembl.org/index.html). The 2,000 bps upstream of the start codons were analysed using the online software PLANTCARE (Lescot et al., 2002). If the distance between two genes was less than 2,000 bps, the sequence between the start codon of the first gene and the termination codon of the second gene were searched for cis-regulatory elements using the PlantCARE database. Additionally, all statistical analyses were performed using the R language (R Core Team, 2019).

Real-time quantitative PCR

To verify the RNA-Seq results, 11 DEGs closely related to endosperm development and starch metabolism were selected for quantitative real-time PCR (qRT-PCR) using SYBR Green I (Bio-Rad) and a CFX96 real-time PCR detection system (Bio-Rad, Hercules, USA). The gene-specific primers used in this assay were designed using Primer Premier 5.0 software (Premier, CAN) and are listed in Table S1, and the specificity of the primers was further verified using the non-redundant (Nr) database and Primer-BLAST (GenBank, NCBI). Each sample from each time point was divided into three biological replicates. Quantification was performed using the comparative CT method with actin (gene ID: GRMZM2G082484) as the internal control. The amplification conditions were 95 °C for 15 min and 40 cycles of 95 °C for 10 s, 60 °C for 20 s and 72 °C for 30 s, and these steps were followed by a melting curve programme (95 °C for 10 s followed by an increase from 65 °C to 95 °C in 5-s increments of 0.5 °C) to confirm the specificity of the PCR amplification. The relative expression levels were calculated as previously described (Livak & Schmittgen, 2001).

Isolation of starch and determination of the amylose content

To eliminate interference from other ingredient, fat was removed using diethyl ether prior to the isolation of starch, and 200 g of corn flour was then steeped in 1,000 mL of distilled water. The mixture was evenly stirred, and the homogenate was passed three times through a high-precision stainless-steel sieve (150 mu/0.106 mm) to improve the purity of the isolated starch. The slurries were then poured into 50-mL centrifuge tubes and centrifuged at 2,000 rpm for 20 min. This procedure was repeated three times. The supernatant in the centrifuge tube was then discarded, and the sediment was suspended in 40 mL of 0.5% (w/v) sodium hydroxide solution for at least 4 h. The supernatant was discarded, and 40 mL of new 0.5% (w/v) sodium hydroxide solution was added. After the mixture was evenly stirred, the centrifuge tube was centrifuged at 4,000 rpm for 10 min. These steps were repeated three times. Subsequently, 30 mL of absolute ethanol was added to the centrifuge tube, and the tube was centrifuged at 4,000 rpm for 10 min. The final sediment was then oven-dried (40 °C) or air-dried and ground using a mortar and pestle. The amylose content of the B73 starch was determined from the size-exclusion chromatography weight distributions of the debranched starch and was calculated as the ratio of the area under the curve of the amylose branches (defined to have a degree of polymerization P 100) to the area under the curve of the entire distribution (including both the amylopectin and amylopectin branches) (Fitzgerald et al., 2009). The amylose content of the sequenced samples (SD609 and HS68) was determined using a two-wavelength iodine binding procedure, which was previously described in detail (Zhu et al., 2008). Two amylose determination methods were used to ensure high-accuracy measurements.

Results

Overview of the RNA-Seq datasets

To obtain global transcriptome maps of the early endosperms of maize with different amylose contents, we performed an RNA-Seq analysis of the common maize SD609 (average amylose content of 27.70%) and the high-amylose maize HS68 (average amylose content of 50.76%) (Fig. 1A). Specifically, the Illumina HiSeq2000 platform was used to perform paired-end RNA-Seq analyses of the endosperm tissues from the two maize plants at 5, 10, 15 and 20 DAP. We generated approximately 0.4 billion high-quality reads, and 95.34% of the mapped reads could be uniquely mapped to the B73 reference genome (Table S2). Additionally, the visualizations of the reads mapped to the maize genome sequences varied across the four developmental stages, which indicated that the transcriptome is dynamic during the early stage of maize endosperm development and that the RNA-Seq data were reliable for further analysis (Fig. S1).

Figure 1 Overview of the amylose content and distribution of the expressed genes.

(A) Amylose contents of SD609, HS68 and B73. (B–I)Distribution of transcripts at five expression levels based on the FPKM values. The pie charts show the number and percentages of transcripts at different expression levels in the SD609 (B-E) and HS68 (F–I) endosperms at different developmental stages. (J–K) Upset plot showing the number of expressed genes that were detected in SD609 and HS68 endosperms among the different development stages.

Dynamic gene expression patterns and cellular function

To gain insight into the temporal and spatial gene expression patterns during the early stage of maize endosperm development, the gene expression levels at each stage of endosperm development in SD609 and HS68 were classified into five categories based on their FPKM values (Fig. 1B, Table S3). The highest level had the lowest number of expressed genes, i.e., approximately 2,000, whereas the lowest level had more expressed genes than the other levels, i.e., approximately 8,000 (Fig. 1). In particular, the difference in the overall number of expressed genes between SD609 and HS68 was only 326, and 22,494 and 22,358 genes were expressed in the endosperm of B73 at 10 and 20 DAP, respectively. Approximately 6,000 extended (expressed) genes were detected in SD609 and HS68 at 10 (29,205 and 28,421, respectively) and 20 DAP (28,723 and 28,144, respectively) (Chen et al., 2014).

To further understand the divergence in expressed genes between SD609 and HS68 and reduce the influence of transcription noise, only genes with FPKM values ≥ 1 were used in the estimation of the number of expressed genes. In total, 21,986 expressed genes were found in the SD609 and HS68 endosperms at all four early stages of development (Fig. 1C), and a GO enrichment analysis suggested that these common genes were mainly involved in cellular carbohydrate biosynthesis, cellular protein metabolism, cellular nitrogen compound biosynthesis, and other essential processes of endosperm development (Fig. S2). A total of 390 genes were expressed only during specific stages of endosperm development in SD609 and HS68, and most of these genes were found to be related to the response to stress and the regulation of gene expression and cell development (Table S4). In particular, sugary enhancer 1 (se1; GRMZM5G842214), which is upstream of the starch metabolism pathway, was only detected at 5 DAP in HS68. However, previous genetic studies have uncovered that se1 is a recessive modifier of su1 that leads to a higher sugar content and an increased kernel moisture for longer post-harvest periods (Ferguson, Rhodes & Dickinson, 1978; Carey, Rhodes & Dickinson, 1982). In addition, increased maltose concentrations have also been detected in mature se1 kernels (Ferguson, Dickinson & Rhodes, 1979; Tadmor et al., 1995). These characteristics indicate that se1 might be a regulator that affects the ratio of amylose to amylopectin in the early stage of endosperm development in HS68.

Overview of gene expression patterns

To understand the gene expression patterns in maize with different amylose contents, pairwise comparisons of all developmental stages of SD609 and HS68 were performed. After genes with read counts lower than 10 at each stage were filtered out, a total of 15,922 DEGs were identified based on the criteria log2fold change (log2FC) ≥1 or ≤-1 and FDR ≤ 0.05. Of these, 1,193 DEGs were obtained from the comparison between SD609 and HS68 at the various developmental stages, 7,813 DEGs were consistently expressed in SD609 and HS68 at the different developmental stages, and 4,914 and 2,002 DEGs were DEGs that were only detected in SD609 and HS68, respectively, and are thus considered material-specific DEGs (Fig. S3, Table S5). A functional enrichment analysis indicated that 1,193 DEGs between SD609 and HS68 were mainly involved in regulation of the intracellular signalling cascade, response to hormone stimulus, system development and fatty acid biosynthetic process (Fig. S4A). In addition, the material-specific DEGs were also found to regulate the same biological processes, such as the intracellular signalling cascade, fatty acid biosynthetic process, dephosphorylation, and response to external stimuli (Fig. S4B, Table S6). Notably, only the material-specific DEGs detected in HS68 were found to be involved in the response to water deprivation, and these DEGs in HS68, which can potentially be attributed to the higher amylose content of this maize variety compared with that of common maize, likely induced the loose binding of starch grains and ultimately resulted in rapid water dispersal, as observed in these kernels at different stages of development but not in those of common maize (Kibar, Gönenç & Us, 2010; Ai & Jane, 2018) (Figs. S4C–S4D). In contrast, the material-specific DEGs of SD609 were enriched in diverse biological processes, such as cell communication, the defence response, nucleosome assembly and the response to oxidative stress (Fig. S4D). These results indicated the existence of obvious metabolic differences between common and high-amylose maize at the early stage of kernel development.

Figure 2 Expression patterns of shared DEGs between SD609 and HS68.

(A–P) Sixteen clusters were characterized by the fluctuating expression of the gene sets at 5, 10, 15 and 20 DAP in both SD609 and HS68. The up- and downregulated gene sets are staggered or depicted consecutively during the development of the maize endosperm, and the same DEG sets with the same or different expression patterns in two different maize materials are shown. The scaled expression levels of the DEGs are provided on the y-axis, the developmental stages are shown on the x-axis, the coloured lines represent the individual gene expression clusters, and the trend in the expression of each gene set is depicted by a black line. “n” represents the number of DEGs.

Figure 3 Analysis of the expression patterns of candidate genes by qRT-PCR and RNA-Seq.

(A, C, E, G, I, K, M, O, Q, S and U) The scale on the left corresponding to the box plots shows the relative gene expression levels based on the qRT-PCR results. (B, D, F, H, J, L, N, P, R, T and V) The scale on the right corresponding to the line charts indicates the gene expression level based on the RNA-Seq results. The x-axis indicates the day of endosperm sampling after pollination. The letters correspond to the genes. The red lines correspond to SD609, and the blue lines correspond to HS68.

A temporal expression pattern analysis of the shared DEGs between SD609 and HS68 was conducted based on k-means clustering and the CH index, and 16 clusters were identified. The results showed that although the same genes were differentially expressed in the two maize materials, their expression patterns were not synchronized and showed a single or multiple transition points at the different development stages. For instance, clusters 6, 11 and 15 exhibited significant differences in expression transition (Fig. 2). In addition, some clusters, such as clusters 7, 8 and 12, showed synchronous fluctuations in their expression patterns but differences in their expression levels (Fig. 2). The qRT-PCR patterns for all candidate genes were almost equivalent to the RNA-Seq patterns (Fig. 3, Table S7). A stringent GO term enrichment analysis (P < 0.0001, FDR < 0.01) of the shared DEGs revealed that the gene sets of most clusters were involved in the regulation of system development, reproductive process, and multicellular organismal development (Fig. S5A, Table S6). Additionally, some clusters were highly correlated with specific biological processes, such as negative regulation of transcription (cluster 5), glucan metabolic process (cluster 6), regulation of catalytic activity (cluster 11) and nucleosome metabolic process (cluster 16) (Fig. S5A).

Figure 4 Sucrose and starch biosynthetic pathways in non-photosynthetic cells.

In non-photosynthetic cells, sucrose is degraded via hydrolysis into hexoses (glucose and fructose) through a reaction catalysed by invertase (INV) or reversibly converted into fructose and uridine diphosphate glucose (UDPG) through a reaction catalysed by sucrose synthase (SUS). Fructose is transformed into frutcose-6-phosphate (fructose-6-P) by fructokinase (FRK) and is further metabolized to G-6-P by glucose-6-phosphate isomerase (PGI). Glucose is then transformed into glucose-6-phosphate (G-6-P) by hexokinase (HXK) and translocated into the amyloplasts via phosphate translators. UDPG is further metabolized to glucose-1-phosphate (G-1-P) by the action of UDPG pyrophosphorylase (UGPase). G-1-P, which can also be obtained from the phosphoglucomutase (PGM)-mediated transformation of G-6-P, serves as a precursor for the ADPG pyrophosphorylase (AGPase)-catalysed formation of adenosine diphosphate glucose (ADPG). Both G-1-P and G-6-P are translocated into the amyloplasts via phosphate translators, whereas ADPG is translocated via ADPG transporters. The content of ADP-glucose is also negatively regulated by Nudix hydrolases (NUDTs), which break down ADP-glucose linked to starch biosynthesis, and starch biosynthesis subsequently occurs in the amyloplast. Starch can be chemically classified into two homopolymers: amylose and amylopectin. Amylose is an almost linear α-1,4 glucan molecule synthesized by AGPase and granule-bound starch synthase (GBSS), whereas amylopectin is a highly branched glucan obtained through a coordinated series of enzymatic reactions involving AGPase, soluble starch synthase (SSS), starch branching enzyme (SBE), and starch debranching enzyme (DBE) (Qu et al., 2018). In particular, GBSS is mainly responsible for the synthesis of amylose and long amylopectin chains. Enzymes that catalyse specific reactions are shown in italics. The heat map shows the expression levels of genes encoding the corresponding enzymes, and the scale bar represents the gene expression value (log10 (FPKM +1)).

We further classified the material-specific DEGs and found no unique expression transition pattern for a specific material. A thorough analysis of the functional enrichment of the material-specific DEGs revealed that some unique expression patterns were closely related to key biological processes in endosperm development. For instance, cluster 9 of SD609 and cluster 3 of HS68, both of which are related to cell differentiation, showed the same expression patterns, and clusters 3 and 8 of SD609 and clusters 1 and 10 of HS68, all of which are involved in the regulation of organ development, exhibited synchronous expression transitions. In addition, cluster 4 of SD609 and cluster 2 of HS68, which are related to negative regulation of biological process, also showed the same expression pattern (Fig. S5–S6, Table S6). Interestingly, we found opposite expression patterns for clusters that are relevant to the same biological process; for example, clusters 3 and 4 of SD609, which are related to negative regulation of biological process, exhibited the opposite expression patterns (Figs. S5–S6). A similar finding was also found for some DEGs between the two different materials; for example, cluster 9 of SD609 and cluster 3 of HS68, which are involved in the regulation of growth, presented opposite expression transitions (Figs. S5–S6). These results indicated that our gene expression pattern sets provide a sufficient dynamic atlas for the identification of genes that function in key biological processes during early maize endosperm development and provide valuable insights for annotating the function of putative genes.

Identification of weighted gene coexpression network analysis (WGCNA) modules associated with starch synthesis-related enzymes

Starch metabolism is a complex physiological process due to its close connection with sucrose metabolism and its regulation by dozens of enzymes (Fig. 4). Additionally, some key enzymes play crucial roles in regulating the synthesis of amylose and amylopectin and might also be crucial factors responsible for the differences in the amylose content between SD609 and HS68. To obtain an in-depth understanding of the potential regulatory mechanisms and core gene networks responsible for the differences in the amylose content between SD609 and HS68, we performed a WGCNA using the 15,922 above-described DEGs and identified 31 WGCNA modules (Fig. 5). We searched the core genes involved in starch synthesis in all the modules based on a previous study (Qu et al., 2018). In total, 19 of the 28 core genes associated with starch synthesis were differentially expressed and were distributed among 10 modules, namely, Plum (AGPLS1), Plum_1 (SSIIa), Steel_blue (AGPLS4), Blue (SBEIIb and SSIV), Dark_magenta (su1), Dark_orange (SBEI, SBEIII and SSIIIb-b), Dark_violet (SSIIb), Grey_60 (AGPLS2, AGPSS3, PUL and SSI), Maroon (SSIIIb-a) and Orange-red_3 (AGPSS1, GBSSI, GBSSII and SSIIIa) (Fig. 6A).

A GO enrichment analysis revealed that genes in each of the 10 modules were not only significantly enriched in specific biological processes but also involved in the regulation of common pathways. For example, the Orange-red_3 module includes the most critical enzyme, GBSS, which regulates amylose synthesis, and other genes belonging to this module were specifically and significantly enriched in the lipid biosynthetic process, fatty acid biosynthetic process, and cell proliferation (Fig. 6B, Table S8). Therefore, the Orange-red_3 module might play an important role in regulating the balance between cell development and the synthesis of nutrients. The Grey_60 module also showed specific enrichment and appeared to be highly associated with the glycerolipid metabolic process, glycerophospholipid metabolic process and phospholipid metabolic process (Fig. 6B, Table S8). These biological processes and the starch biosynthesis process both utilize fructose-6-P as a substrate for further extension, which suggests that the Grey_60 module likely affects the relationships between sucrose decomposition and starch synthesis through substrate competitive inhibition. Additionally, it should be noted that the 10 modules exhibited a high correlation and formed a cascade-based community, and a functional analysis revealed that genes in the 10 modules were highly significantly enriched in the reproductive process (Fig. 6B, Fig. S7). This result indicated that the reproduction process might be the most fundamental biological process and could further affect other essential metabolism processes, including the starch synthesis process, by interacting with multiple genes belonging to different modules during the early stages of maize endosperm development.

Figure 5 Modules from weighted gene coexpression network analysis (WGCNA).

Each module was given an arbitrary colour-based name and is summarized by a metric known as the module eigengene (ME), which is the first principal component of the module (i.e., the axis capturing the majority of the variation in expression in the module). The relationships among the modules are shown in the dendrogram and heat map. The values are based on the linear regression coefficients of determination, which were obtained using the module eigengene values from the WGCNA analysis. Higher regression scores indicate more similar expression patterns between the modules or an increased predictive power of the main effects on module expression in the dendrogram (closer clustering) and heat map (deeper red).

Identification of hub genes and key networks associated with starch synthesis

Most maize agronomic traits and metabolic processes are complex due to not only the actions of single genes but also interactions between combinations of genes. A sizeable portion of the genes in each network module exhibited extremely high connectivity with other genes belonging to the 10 modules in which the core genes in starch synthesis were distributed. These highly connected genes are usually considered hub components of a network and have been shown to be biologically important in multiple previous applications. Therefore, we performed a NetworkAnalyzer analysis and found that 5% (918) of the genes in the 10 modules had a higher degree of connection and could thus be identified as hub genes for further study (Table S9). A functional enrichment analysis of these hub genes indicated that many of these genes encode proteins functioning in chromatin and protein binding, oxidoreductase activity and transferase activity (transferring hexosyl and glycosyl groups). In addition, these hub genes mainly play an important role in the response to hormone stimulus, regulation of the cellular component size, system development, heterocycle catabolic process, and response to water deprivation (Table S9).

Figure 6 Relationship and functional annotation of the modules.

(A) Relationships between different modules. Only the correlation values between different modules that are greater than 0.5 are shown. (B) KEGG enrichment analysis of genes in all the modules. Only metabolic processes with a P value < 0.05 are shown.

According to the gene classification results, 77 transcription factors of 918 hub genes were identified, and these belonged to distinct families, some of which (AP2/ERF, ARF, bHLH, MYB, WRKY, NAC, bZIP, and GRAS) have been reported to play roles in cell development and the synthesis and accumulation of starch in maize, rice, hulless barley, Arabidopsis, and other plants (Iwase et al., 2011; Licausi, Ohme-Takagi & Perata, 2013; Liu et al., 2015; Huang et al., 2016; Butardo Jr et al., 2017; Cai et al., 2017; Tang et al., 2017; Xiao et al., 2018) (Table S9). In particular, only one core gene of starch synthesis (GRMZM2G158043, ZmPUL) was identified as a hub gene in the Grey_60 module, and this gene is mainly involved in the modification of excessively branched chains or the removal of improper branches of amylopectin formed by branching enzymes to maintain the cluster structure of amylopectin. More remarkably, beta-glucosidase 18 and aldose 1-epimerase (encoded by GRMZM2G100452 and GRMZM2G031628, respectively) were also found in the Grey_60 module and are mainly related to carbohydrate transport and metabolism (Table S8). These results indicated that some hub genes of the Grey_60 module might play a greater role in regulating the synthesis of amylose and amylopectin. However, this finding raises the question of whether other core genes of starch biosynthesis are key factors that determine the contents of amylose and amylopectin.

To further explore the roles of core genes of the starch synthesis pathway in regulating the contents of amylose and amylopectin, we extracted the coexpression network of 19 differentially expressed core genes of starch synthesis. The analysis revealed that starch biosynthesis is regulated by multiple cascade networks that fragment into more sub-networks, and some of the sub-networks of the core genes of starch synthesis, particularly the sub-networks centred on ZmPUL, were linked by one or more linking genes (Fig. 7). Other disjoint sub-networks might rely on more linking genes as bridges in the regulation of starch synthesis (Fig. 7). Additionally, the relationships of the genes in the community network were also further supported by those of the protein-protein interactions in the STRING database (https://string-db.org/) presented in a previous study (Qu et al., 2018). A rigorous functional enrichment analysis indicated that these linking genes play crucial roles in cell cycle regulation, replication, the MAPK cascade, energy metabolism, binding (including protein binding, nucleotide binding and ion binding), carbohydrate transport and metabolism, lipid transport and metabolic processes, amino acid transport and metabolism, and translation and posttranslational modification (Table S10). These results further indicated that the above-mentioned biological processes might be intimately linked to starch synthesis, and the specific functions of core genes of starch synthesis might be affected by regulatory genes of these metabolism processes. Nevertheless, the detailed regulatory and interactive mechanisms between core genes of starch synthesis and genes involved in other key biological processes remain to be elucidated.

Figure 7 Starch biosynthesis community networks.

The nodes correspond to genes, and the edges represent coexpression links. Only those with a weight value greater than 0.5 are shown. These networks include 2804 DEGs and were subdivided into 19 core gene modules based on the key genes of starch synthesis, and these modules are marked with a red colour and the corresponding gene names. Each core gene module is shown with different background colours, and the interactive genes in different modules are distinguished by different colours. The nodes between core gene modules are marked with a pink colour.

Discussion

As the major storage carbohydrate in plants, starch has attracted considerable attention as a significant food ingredient and biomaterial. Different contents of amylose and amylopectin in starch give rise to important differences in the properties and functionality of starches obtained from different plants. Nevertheless, only a few studies have examined the differences between corn materials with different amylose contents at the level of transcription. In this study, we performed in-depth transcriptomic surveys based on the high-throughput RNA sequencing of samples of endosperm tissue from the common maize SD609 and the high-amylose maize HS68 collected at various times. Our results showed that at least 28,378 and 28,144 genes are required for the developmental programming of the respective maize endosperms. In addition, a comparative analysis revealed that the difference in the overall number of expressed genes between SD609 and HS68 was only 326. These results will facilitate the identification of gene regulatory networks and essential metabolic processes that are important for the programming of endosperm development.

Confirmation of the phases of different processes is important for expounding the endosperm developmental stages. Gene expression pattern analysis is an effective tool to depict a powerful atlas for uncovering the key players and the complexity of different processes during stages of endosperm development. An expression pattern analysis of the specific DEGs of SD609 and HS68 in our dataset suggested that most of the temporal patterns were similar and that the only difference was in the expression level (Fig. S6). However, obvious differences in the transition points and expression patterns of the shared DEGs of SD609 and HS68 appeared at 10 and 15 DAP (Fig. 2). This result indicated that some developmental processes might not be synchronous between SD609 and HS68, and the biological functions of some synchronous or asynchronous expression clusters of SD609 and HS68 further supported this hypothesis. Additionally, some synchronous expression pattern clusters between SD609 and HS68 were found to main functioned in cell differentiation, multicellular organismal development, nucleosome metabolic processes and glucan metabolic processes (Fig. S5A, Table S6). These results agree with the morphogenetic characteristics of the endosperm. For instance, the endosperm grows rapidly from 8 to 12 DAP until it fills the entire seed cavity, and the maize endosperm cells gradually and asynchronously switch from a mitotic to an endoreduplication cell cycle. Genes related to carbohydrate biosynthesis and energy reserve metabolic processes are upregulated at 10 DAP. Cell division continues until approximately 20 to 25 DAP in the external cell layer of the endosperm, and this layer then develops into the aleurone and subaleurone layers (Sabelli & Larkins, 2009; Leroux et al., 2014; Li et al., 2014).

The starch metabolism process is a metabolic process mainly based on energy reserves. In addition, starch biosynthesis is a complex and highly regulated process that requires coordinated activities among multiple enzymes, and different isozymes exert different effects (Jeon et al., 2010; Qu et al., 2018). We analysed the core genes of starch synthesis and found that the main differences in the expression patterns of 28 core genes between SD609 and HS68 were observed at 10 and 15 DAP, and the expression transition patterns of some core genes agreed with the reported expression patterns (Fig. 8A) (Li et al., 2014). For example, GBSSI, a crucial enzyme that regulates the synthesis of amylose, showed higher expression levels at 10 (one step up) and 15 DAP (one step down) in HS68 compared with SD609, and this upregulated pattern agreed with that in maize B73 at 10 DAP and was further verified by qRT-PCR (Figs. 3 and 8). In addition, other core genes, such as SBEIIb, AGPLS1 and AGPSS3, also shared a similar temporal pattern (downregulated-downregulated-upregulated), and the expression levels of AGPLS1 and AGPSS3 were higher in HS68 than in SD609. This finding indicated that AGPLS1 and AGPSS3 might be the most efficient partners that interact with each other and polymerize into the native heterotetrameric enzyme structure in the high-amylase maize HS68. Although AGPSS also forms a homotetramer, it also shows defective properties in terms of catalysis and regulation, requires a higher concentration of 3-PGA for its activation and is more easily inhibited by Pi (Kavakli et al., 2001; Salamone et al., 2000). However, the expression level of SBEIIb was higher in SD609 than in HS68 at 5 and 10 DAP. In the starch synthesis process, SBEII contains two isoforms, SBEIIa and SBEIIb, which play important roles in promoting the production of the short amylopectin chain (degree of polymerization of 6-12 chains) and further impact the structure and phenotype of amylopectin, but the differences in the expression of SBEIIb between HS68 and SD609 indicated that SBEIIb might be induced by increasing the branch frequency and lengthening the glucan chains in clusters to improve the ratio of amylopectin (Fig. 8A) (Nishi et al., 2001; Liu et al., 2009).

Figure 8 Expression map of core genes of starch synthesis and zein genes.

(A–B) Heat map showing the expression patterns of core genes of starch synthesis (A) and zein genes (B) in the SD609 and HS68 endosperms at different developmental stages. The scale bar shows the normalized FPKM values.

We measured the amylose content in various maize materials and found that the materials could be ordered as followed from a high to a low amylose level: HS68 (50.76%) >SD609 (27.70%) >B73 (25.16%) (Fig. 1A). The combination of this information with the results from the expression analysis of the core genes of starch synthesis revealed that the amylose content of different maize materials might be affected by the expression levels of certain core genes. Additionally, the synthesis patterns of amylose and amylopectin might be regulated by a multienzyme complex (Fig. 5). For example, SSI, SSIIa, SBEI and SBEIIb showed the same expression pattern, which is consistent with the results of earlier studies that showed the following: the SSI and SSII isoforms and either SBEIIa or SBEIIb form a trimeric complex to regulate the starch metabolic pattern (Liu et al., 2012). Furthermore, SBEIIb deficiencies have been shown to affect the binding of SSI and SBEI to starch granules in the amylose extender (ae1.1) mutant, and an experiment involving mutations in the amylose extender (ae1.2) also showed that SSI, SSIIa, SBEI and SBEIIb form another multiple-enzyme complex (Liu et al., 2009; Abe et al., 2014). In particular, obvious differences in the amylose content and granule size have been described between two ae mutations from two near-isogenic maize lines (Liu et al., 2012).

In addition to the above-described factors that affect starch synthesis, we revealed that each differentially expressed core gene involved in starch synthesis, including members of multiple TF families, can be classified into a potential interaction module containing multiple genes (Fig. 7). For example, ZmEREB156 (GRMZM2G421033), a member of the AP2/EREBP TF family that positively modulates the starch biosynthetic gene SSIIIa via the synergistic effect of sucrose and ABA, exhibited two transition points (with the upregulated-downregulated-upregulated pattern) in SD609 and HS68 (Huang et al., 2016). Consistent with the above-described features, one plant_AP-2-like regulatory element and seven ABA response elements were found in the promoter regions of core genes of starch biosynthesis (Fig. 9). In addition, the identified cis-regulatory elements also include various response elements, such as multiple hormone response elements (i.e., SA, JA, auxin, ETH and GA), stress response elements, and cell development response elements (Fig. 9). GRAS20 (GRMZM2G023872) belongs to the GRAS TF family and can reduce fractions of long-branched starch chains, and its overexpression leads to the formation of a chalky region in the ventral endosperm with a decreased starch content. In addition, compared with wild-type plants, GRAS20-transgenic seeds exhibit an altered starch granule morphology, which leads to defective agronomic characteristics (Cai et al., 2017). In our study, GRAS20 showed different expression patterns in the two maize materials with different amylose contents, i.e., upregulated-downregulated-upregulated (SD609) and upregulated-upregulated-upregulated (HS68), which suggested that GRAS20 might be involved in the regulation of amylose synthesis. Moreover, some other members of the TF family have also been identified to play a role in the regulation of starch synthesis, but these were not detected in the core gene modules of starch synthesis. For example, O2, an endosperm-specific bZIP TF that affects the protein levels of SSIIa, SSIII and SBEI, showed a high expression level and exhibited consistent expression patterns between the RNA-Seq and qRT-PCR results obtained in our study but was not present in the core gene modules (Zhang et al., 2016). One possible reason that explains the absence of O2 is that it might affect other gene(s) to regulate starch synthesis, which is in line with the results of a previous study (Zhang et al., 2016); alternatively, O2 might have been replaced by other bZIP TF(s) members because five bZIP TFs were detected in the core gene modules of starch synthesis (Table S11). However, the results of the analysis of cis-regulatory elements in the promoter regions of core genes of starch synthesis indicate that the first possibility is more likely because an O2-binding site was found in the vast majority of core genes of starch synthesis (Fig. 9).

Figure 9 Cis-regulatory elements in promoter regions of core genes of starch synthesis.

(A–R) Heat map showing the number of regulatory elements identified in core genes of starch synthesis. The scale bar corresponds to the left side, which shows the name and classification of regulatory elements according to the PlantCARE database. The right side shows the sequence information corresponding to the element. The scale bar indicates the range of element numbers. Abbreviations: abscisic acid (ABA), salicylic acid (SA), jasmonic acid (JA), ethylene (ETH), and gibberellin (GA).

Zeins are abundant storage proteins in most cereals and constitute an important factor that affects seed quality. According to their structural differences, zeins are subdivided into four subclasses: α-zeins (considered the major subclass, with 71–85% of total zeins), γ-zeins (10–20%), β-zeins (1–5%) and δ-zeins (1–5%) (Esen, 1987; Xu & Messing, 2008). In our study, we detected 30 zein genes, including 25 α-zeins, one β-zein, three γ-zeins and one δ-zein, and most of these showed a high expression level in SD609 and HS68 at the early stage of endosperm development (Fig. 8B). A comparison of our results with the gene expression sets obtained for maize B73 revealed obvious differences in the number of expressed zein genes and expression patterns among SD609, HS68 and B73 (Chen et al., 2014). Thus, we speculate that β-, γ-, and δ-zeins and some α-zeins are likely essential zeins during early maize endosperm development. Previous studies support the hypothesis that the smallest zein-containing protein bodies, such as β- and γ-zeins, are observed in subaleurone cells and play a role in α- and δ-zein retention in the rough endoplasmic reticulum, whereas α-zeins and some δ-zeins have been detected in subaleurone and starchy endosperm cells (Coleman et al., 1996; Bagga et al., 1997; Woo et al., 2001). Additionally, an analysis of the interactions between zeins revealed that 50-, 27-, and 16-kD γ-zeins and 15-kD β-zein exhibit strong interactions, and strong interactions have also been detected between α- and δ-zeins and the 16-kD γ-zein and the 15-kD β-zein, respectively (Kim et al., 2002). Moreover, many studies have noted that all zein classes must be present at correct stoichiometric ratios to ensure proper formation of the protein body and the resulting vitreous endosperm, and specific zein mutations during zein synthesis not only alter the protein body shape and size but also influence the endosperm texture and cause opacity (Guo et al., 2013; Holding, 2014; Holding et al., 2007; Kim et al., 2004; Wang et al., 2011; Wang et al., 2012). These results further suggested that the expressional differences in α-, β-, γ- and δ-zein-encoding genes are likely to affect the development of the maize endosperm. Interestingly, previous studies have revealed that O2 encodes and mainly regulates the expression of α- and β-zein genes by recognizing the O2 box in their promoters. Furthermore, O2-heterodimerizing proteins, O2 and the prolamine box-binding factor are master regulators of zein synthesis by acting in an additive and synergistic mode (Schmidt et al., 1990; Schmidt et al., 1992; Zhang, Yang & Wu, 2015). Additionally, it is worth noting that a recent study revealed that O2 and the prolamine box-binding factor also affect the protein levels of SSIIa, SSIII and SBEI (Zhang et al., 2016), which implies that O2 might be a bridge that connects starch synthesis and zein synthesis or could potentially regulate the levels of other TF(s) or mRNA regulatory factor(s) and thereby affect the functions of the core genes involved in starch synthesis and zein genes. Nonetheless, the detailed regulatory mechanism through which zein proteins affect amylose/amylopectin biosynthesis remains to be determined, but this study provides key transcriptional-level clues that will aid further elucidation of the regulatory relationships between amylose biosynthesis and zein-encoding genes.

Conclusions

In summary, this study offers essential information for identifying the gene atlas of common and high-amylose maize and further distinguishing the expression patterns of genes in the endosperm of common and high-amylose maize. In addition, this analysis provides new clues for exploring the regulatory networks of starch synthesis and will help improve our understanding of endosperm development and the mechanism underlying the regulation of starch synthesis. It also provides valuable information resources for the breeding of maize materials with different starch contents.

Supplemental Information

Figure S1 Read coverage of the SD609 and HS68 endosperms at the four developmental stages

The reads at 5, 10, 15 and 20 DAP mapped to maize reference genome sequences are shown from the outside to the inside of the Circos image. The red and green colours represent the normalized read numbers of SD609 and HS68, respectively.

Click here for additional data file.

Figure S2 Functional enrichment analysis of the shared genes among the four developmental stages in the SD609 and HS68 endosperms

(A) The enriched GO terms in the “biological process”, (B) “cellular component”, and (C) “molecular function” categories are shown.

Click here for additional data file.

Figure S3 Number of DEGs identified in SD609 and HS68

A total of 1,193 DEGs were differentially expressed between SD609 and HS68, 7,813 DEGs were consistently expressed in SD609 and HS68 at the different development stages, and 4,914 and 2,002 DEGs were specifically detected in SD609 and HS68, respectively.

Click here for additional data file.

Figure S4 Functional enrichment analysis of DEGs at different developmental stages

(A) Functional annotation of 1,193 DEGs between SD609 and HS68. (B) Shared biological processes regulated by the material-specific DEGs of SD609 and HS68. (C) Specific biological processes of HS68 that are regulated by the material-specific DEGs of SD609 and HS68. (D) Specific biological processes of SD609 that are regulated by the material-specific DEGs of SD609 and HS68. Only biological processes with P values < 0.0001 and FDR < 0.01 are shown.

Click here for additional data file.

Figure S5 Functional enrichment analysis of the DEG sets of the coexpression clusters

Only biological processes with a P value < 0.0001 and FDR < 0.01 are shown. (B) Functional analysis of coexpression clusters of SD609. (C) Functional analysis of coexpression clusters of HS68. (A) Functional analysis of shared DEGs in the coexpression clusters between SD609 and HS68.

Click here for additional data file.

Figure S6 Expression patterns of the DEGs in the developing maize endosperm

(A) Twelve clusters were characterized by the fluctuating expression of gene sets at 5, 10, 15 and 20 DAP in SD609. (B) Eleven clusters were characterized by the fluctuating expression of gene sets at 5, 10, 15 and 20 DAP in HS68. The up- and downregulated gene sets are staggered or depicted consecutively during the development of the maize endosperm. The scaled expression levels of the DEGs are provided on the y-axis, the developmental stages are shown on the x-axis, the coloured lines represent the individual gene expression clusters, and the trend in the expression of each gene set is depicted by a black line. “n” represents the number of DEGs.

Click here for additional data file.

Figure S7 Relationship between module eigengenes

The diagonals show the distribution. The lower left section shows a bivariate scatterplot with a fitting line, and the upper right section shows the correlation coefficient and the significance level.

Click here for additional data file.

Table S1 List of primers used for gene detection by qRT-PCR analysis

Click here for additional data file.

Table S2 Statistical analysis of the sequenced and mapped reads of SD609 and HS68

Click here for additional data file.

Table S3 Information on the expressed genes of SD609 and HS68

Click here for additional data file.

Table S4 Functional enrichment analysis of stage-specific gene expression

Click here for additional data file.

Table S5 Information on DEGs of SD609 and HS68

Click here for additional data file.

Table S6 GO enrichment analysis of the DEGs of SD609 and HS68

Click here for additional data file.

Table S7 qRT-PCR analysis of candidate genes

Click here for additional data file.

Table S8 Functional annotation of the WGCNA modules

Click here for additional data file.

Table S9 Functional annotation of the hub genes

Click here for additional data file.

Table S10 Core gene networks of starch biosynthesis

Click here for additional data file.

Additional Information and Declarations

Competing Interests

Author Contributions

Data Availability

The authors declare there are no competing interests.

Jianzhou Qu performed the experiments, analyzed the data, contributed reagents/materials/analysis tools, prepared figures and/or tables, authored or reviewed drafts of the paper.

Shutu Xu, Xiaokang Tian, Ting Li, Licheng Wang and Yuyue Zhong approved the final draft.

Jiquan Xue and Dongwei Guo conceived and designed the experiments.

The following information was supplied regarding data availability:

The data is available at the National Center for Biotechnology Information Sequence Read Archive: SRP065059 and SRP149609.

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
