# Peer review of "Comparative transcriptomics reveals the difference in early endosperm development between maize with different amylose contents"

_PeerJ, doi:10.7717/peerj.7528_

## Round 0.1 · original submission · Major Revisions

I have received favorable recommendation from two independent reviewers, who do not have any conflict of interest with your work. Both have pointed out several concerns which can be addressed at your end. Please address to all of their comments in a point-wise manner.

Also, the language used in the manuscript needs to be polished and improved by the authors for an international audience, by the help of native English speaking colleagues or other means. Without a thorough language proof and quality, I might not be able to accept the work.

Further, I have some comments and please address these:

Figure 1.(C): The 4-way Venn diagram is far more confusing than conveying any message. Please use UpsetR: https://github.com/hms-dbmi/UpSetR to show those shared and unique DEGs which is way informative and easy to do.
Figure 3. Present as Box-Whisker plots for showing the data point spread, means etc.
Figure 5. Which of these modules are statistically significant? Rather than random chance. Need to be specified.
Figure 6. Instead of or in addition to GO ontology, provide 'Pathway enrichments' which would be more relevant and informative for each module, rather than GO.
Figure 8. A, B need to be specified. Rather than heatmaps side by side, can not show scatter plots of zein and starch sysnthesis core genes?

·

Basic reporting

No comment.

Experimental design

Enhance description of the research question (end of Introduction).

Validity of the findings

Enhance concluding remarks at end of the manuscript.

Additional comments

The current manuscript “Comparative transcriptomics reveals the difference in early endosperm development between maize with different amylose contents” is a solid work that compares two maize lines with different amylose content at different stages of endosperm development through transcriptomics analysis. It provides valuable data for research in the area by suggesting key genes and regulatory networks that are fundamental of this stage. It already suggests some interesting ideas on the role of some genes in starch synthesis, but the major work remains to be done – exploration and confirmation of the provided data. It is overall well written, with some exceptions.

I have some comments and then a list of small errors to be corrected.

1) At the end of the Abstract, add some potential benefits that this knowledge (timing of endosperm development) has for agricultural production or any other application. Also clarify your working hypothesis at the end of the Introduction. That includes clarifying what is “other essential metabolisms” (line 120).

2) Line 41: what do you mean with “maize starch is one of the best-quality starches, with a purity that can reach 99.5%”. This may be linked to extraction and purification protocols, which may be well developed for maize but not for other crops. Please clarify.

3) Gene and mutant nomenclature are not consistent nor accurate. Example: line 65 (stands for the rest of the manuscript): globby-1 should be in capital letters, similarly to other gene names. In addition, the gene should be named GLOBBY1, and the mutant globby1-1 (example: Costa et al 2003). Also check for consistency in italics when referring to a gene (example lines 81-82). In addition, make sure to describe names of proteins and their roles throughout the text. Indicate what mutants are you referring to (example: line 104-105)

4) Lines 291-294: clarify in the Results section how the 16 clusters were obtained, and how many genes are included in each cluster.

5) Lines 547-554 consider rephrasing the entire paragraph, it is confusing in some parts and the English clearly needs to be enhanced. Consider as well stating again the agronomic importance/applications of future advancement of this research line.

6) Figure legends:
- figure 1B does not show a “vertical axis”, consider rephrasing
- figure 7: consider describing the different colors and to what genes are they referring to

And then some small details:
- line 26, add “with” after “correlated”
- line 44: I would say endosperm development starts “after” double fertilization, and not “through” as stated now. Please clarify when during plant development the endosperm starts being formed and when is it fully established.
- lines 46 and 50 (stands for the rest of the manuscript): double check text for consistency, example cellularisation VS cellularization
- line 89: state the name of the SSIIa and SBEI proteins
- line 104: add “in” before “nonmutant”
- line 105: clarify if higher levels of amylopectin (in the wx1 mutant) result in lower amylose content
- line 130: consider rephrasing, the way it is stated now it seems that the discarded material was frozen in liquid N2 and used for RNA extraction
- line 189: “Eleven DEGs…” correct the rest of the sentence
- line 234: add “developmental” after “four”
- line 273/274: double check the DEG numbers that are detailed in the text VS the numbers show in Suppl Figure 3 (example SD609 4905 VS 4914)
- line 299: add “were” after “clusters”
- line 485: add “seen/described” after “been”

Reviewer 2 ·

Basic reporting

The present manuscript submitted by the Xue and Guo et al., is an attempt to enhance our understanding of spatiotemporal regulation of starch biosynthesis in maize plant using a comparative transcriptomics approach. Authors selected two contrasting varieties of maize, of which one is having high amylose content (HS68) and the other one has a normal (SD609). The global transcriptome has been carried out at 4 developmental stages day after pollination (DAP 5, 10 15, 20) in both cultivars. As a result, thousands of genes were identified differentially regulated during 4 developmental stages of the two varieties. Of these, authors identified core genes involved in starch synthesis pathway in both maize varieties with limited success. For example, this is surprising that O2, an endosperm-specific TF affects the expression of starch biosynthetic pathway genes (SSIIa, SSIII, SBEI) reportedly shows high expression in transcriptome and qRT-PCR but not present in the core gene group. Overall, the present manuscript is well organized, written and presented. I think this manuscript deemed fit for publication after minor revisions.

First and foremost, I would like to see more information from this huge chunk of transcriptome data. It can be meaningful if authors can perform promoter analysis of clustered genes. What kind of common up-stream regulatory elements are involved in regulating the starch content in HS68 and SD609? This is an important question to know.

There is a discrepancy in no. of genes mentioned in the text (Line no. 248) and shown in supplementary table S3. In table S3, the no. of genes are mentioned as 28144 for HS68 genotype at 20 DAP but in the text, it is 22358 for HS68. Which one is correct?.

What is the FPKM value? Authors should abbreviate in the text at least at the first occurrence (Line no. 241).

Fig. 1C is a bit complicated, I guess it should redraw to make it simple and self-explanatory.

Line no. 269, please mention the name of two specific developmental stages.

Supplementary figure 3 is quite confusing. Authors say 1193 (7%) genes were differentially regulated between SD609 and HS68, what does it mean? In the same figure, 4919 (31%) genes were uniquely expressed in SD609 and 2002 uniquely expressed in HS68. All these genes are differentially expressed. How it (1193) has been derived from the total no. of genes 15922?

Line 271-274, this sentence needs to be rearranged for better clarity.

Line no. 277-278, what does it mean by the “Material-specific DEGs were also found to regulate the same biological process.” What does the word “material-specific” imply?

In table S7, author should remove the clutter by filtering the data on the basis of FDR value (<0.05). By this way, data can be viewed and interpreted in a better manner.

In line no. 282, authors used the word "maize species" I suggest "varieties" or "cultivars" will be the correct word.

Experimental design

For better clarity, the experimental design needs to be illustrated schematically.

Validity of the findings

Altogether, the findings are backed by the data. The data is analyzed appropriately and statistical parameters are robust. I do not see any problem in this section.

Additional comments

The present manuscript submitted by the Xue and Guo et al., is an attempt to enhance our understanding of spatiotemporal regulation of starch biosynthesis in maize plant using a comparative transcriptomics approach. Authors selected two contrasting varieties of maize, of which one is having high amylose content (HS68) and the other one has a normal (SD609). The global transcriptome has been carried out at 4 developmental stages day after pollination (DAP 5, 10 15, 20) in both cultivars. As a result, thousands of genes were identified differentially regulated during 4 developmental stages of the two varieties. Of these, authors identified core genes involved in starch synthesis pathway in both maize varieties with limited success. For example, this is surprising that O2, an endosperm-specific TF affects the expression of starch biosynthetic pathway genes (SSIIa, SSIII, SBEI) reportedly shows high expression in transcriptome and qRT-PCR but not present in the core gene group. Overall, the present manuscript is well organized, written and presented. I think this manuscript deemed fit for publication after minor revisions.

First and foremost, I would like to see more information from this huge chunk of transcriptome data. It can be meaningful if authors can perform promoter analysis of clustered genes. What kind of common up-stream regulatory elements are involved in regulating the starch content in HS68 and SD609? This is an important question to know.

There is a discrepancy in no. of genes mentioned in the text (Line no. 248) and shown in supplementary table S3. In table S3, the no. of genes are mentioned as 28144 for HS68 genotype at 20 DAP but in the text, it is 22358 for HS68. Which one is correct?.

What is the FPKM value? Authors should abbreviate in the text at least at the first occurrence (Line no. 241).

Fig. 1C is a bit complicated, I guess it should redraw to make it simple and self-explanatory.

Line no. 269, please mention the name of two specific developmental stages.

Supplementary figure 3 is quite confusing. Authors say 1193 (7%) genes were differentially regulated between SD609 and HS68, what does it mean? In the same figure, 4919 (31%) genes were uniquely expressed in SD609 and 2002 uniquely expressed in HS68. All these genes are differentially expressed. How it (1193) has been derived from the total no. of genes 15922?

Line 271-274, this sentence needs to be rearranged for better clarity.

Line no. 277-278, what does it mean by the “Material-specific DEGs were also found to regulate the same biological process.” What does the word “material-specific” imply?

In table S7, author should remove the clutter by filtering the data on the basis of FDR value (<0.05). By this way, data can be viewed and interpreted in a better manner.

In line no. 282, authors used the word "maize species" I suggest "varieties" or "cultivars" will be the correct word.

---

## Round 0.2 · accepted · Accept

The manuscript has been deemed suitable for a publication in PeerJ by the peer reviewers.

Reviewer 2 ·

Basic reporting

I am glad to see that the authors have addressed the questions and concerns raised by reviewers and editors sincerely. They have thoroughly revised the manuscript wherever necessary. Authors have replaced the figure 1C by a better explanatory figure. They have added the upstream regulatory element analysis (Fig. 9) that further strengthen the manuscript. The language is revised, factual errors are rectified. At this juncture, the manuscript is deemed fit for the publication.

Experimental design

This section is well written with every detail so that it can be reproduced.

Validity of the findings

Now manuscript is in good shape and findings are clear with regard to high starch biosynthesis related transcriptomic responses.

Additional comments

As above.